# Reimplementing Fairness by Learning Orthogonal Disentangled Representations

**[anonymous double-blind version]**

## 1 Reproducibility Summary

### Scope of Reproducibility

Sarhan et al. [2020] propose a method of learning representations that can be used in downstream tasks, yet that are independent of certain sensitive attributes, such as race or sex. The learned representations can be considered "fair" as they are independent of sensitive attributes. The authors report results on five different datasets, which most notably include (1) the ability of the representations to be used for downstream tasks (target prediction accuracy) and (2) the extent to which sensitive information is present in these representations (sensitive prediction accuracy).

In this text we report and compare the obtained results as well as highlight any difficulties encountered in reproducing the reference paper by Sarhan et al. [2020].

### Methodology

As there was no openly available code base, we re-implemented the work. We included scripts to automatically download the required data, designed the dataloaders, and implemented the models as described in the reference paper. Code is available `https://github.com/paulodder/fact2021`.

### Results

We were able to reproduce some of the results of the paper, but a significant part of our results was inconsistent with the findings of Sarhan et al. [2020]. For some of the simpler datasets we found similar patterns, but for the more complex tasks the models training became unstable, leading to results that varied significantly across random seeds. This made reproduction infeasible.

### What was easy and what was difficult

Conceptually, the paper was interesting and, given some prior knowledge on Variational Autoencoders and the math involved, it was also relatively straightforward to understand.

The most difficult aspect of the project was dealing with missing information. Many essential implementation details were missing, and there were inconsistencies in the pseudo-code provided. Resolving these issues provided significant difficulties.

### Communication with original authors

Various emails were exchanged with the original authors, in which we received explanation about unclear aspects of the paper. In general, the authors were very helpful, but despite the fact that a few emails were exchanged, some aspects of the paper still remained unclear.

Submitted to ML Reproducibility Challenge 2020. Do not distribute.

# 1 Introduction

A challenging problem in machine learning is learning representations that are fair with regards to a sensitive attribute present in the original samples. A common definition of fairness in this context is that the model's output is statistically independent from the sensitive attribute [Xie et al., 2017, Roy and Boddeti, 2019a, Quadrianto et al., 2019, Barocas et al., 2019]. Sarhan et al. [2020] proposed a new way to learn fair representations that are invariant towards the sensitive attribute, but are nevertheless useful for the task at hand. The proposed method to achieve this invariability is to disentangle the latent representation into independent target and sensitive representations [Locatello et al., 2019]. As a proxy for independence, orthogonality is enforced between these individual representations. Furthermore, in order to prevent sensitive information leaking into the target representation, the model is trained to learn a target representation which is agnostic to the sensitive information, maximizing the entropy of our sensitive attribute given the target representation.

In order to consolidate the claims brought forth in the reference work of this report Sarhan et al. [2020], and to assess the reproducibility of this work, our research attempts to reproduce the achieved results by re-implementing the proposed method. In the next section, we specify the parts of the original work that we attempt to reproduce. In Section 3, we summarize the method as proposed by Sarhan et al. [2020] that we attempt to re-implement. In Section 4, we report the results we attain. Finally, in Section 5, we discuss the results, and we analyse the reproducibility of the reference work.

# 2 Scope of reproducibility

The reference work by Sarhan et al. [2020] works towards a method of generating embeddings of data which are useful for downstream tasks, while they remain invariant towards a particular (sensitive) feature. The efficacy of this proposed method is assessed using two evaluative questions. First, how well can the learned target representation be used in the target task? Second, to what degree is information which might reveal the sensitive attribute still present in the learned target representation?

In the reference work, a collection of three binary- and two multi-class classification tasks are considered for a total of five classification tasks, corresponding to five different datasets. For each of these tasks, a version of the proposed model is trained, and evaluated using two metrics: *target accuracy* and *sensitive accuracy*. The target accuracy measures how well a predictive model is able to predict the target attribute based on the produced target representation, and the sensitive accuracy measures how well a predictive model is able to predict the sensitive attribute based on the target representation – note that we want the sensitive accuracy to be as low as possible, because this implies that the target representation is independent of the sensitive attribute. Both of these accuracies, for each of the five tasks at hand, are included in our reproduction.

Furthermore, an ablative study is conducted in the reference work, in which specific components of the loss function used to train the model are excluded (i.e., ablated), in order to observe the behaviour of the model and, in doing so, understand the role of each of these loss components within the training process. This ablative study, which entails the evaluation of the impact of five unique combinations of loss components, is performed on each of the five datasets, and is included in our reproduction.

Finally, the authors perform a sensitivity analysis on the hyperparameters that control the relative importance of two of the loss terms they used, for one of the five tasks. For each combination of these hyperparameters, the model is trained, and the resulting target and sensitive accuracies achieved are displayed on a heatmap. We include this sensitivity analysis in our reproduction.

# 3 Methodology

Since the code of the original implementation is not available, it is our goal to reproduce the method, based on all implementation details expounded in the reference work. The essential elements of the model are described in the next section. For a more detailed explanation, we refer the reader to the reference work Sarhan et al. [2020].

## 3.1 Model descriptions

Let $\mathcal{X}$ be the dataset and let $\boldsymbol{x} \in \mathbb{R}^D$ be a single input sample. Each sample has an associated target vector $\boldsymbol{y} \in \mathbb{R}^n$ and an associated sensitive attribute vector $\boldsymbol{s} \in \mathbb{R}^m$, with $n$ and $m$ classes respectively. The goal is to learn to map $\boldsymbol{x}$ to two latent representations: a target latent representation $\boldsymbol{z}_T$ and a sensitive latent representation $\boldsymbol{z}_S$. This mapping is learned by an encoder, which is composed as follows: the first part of the encoder, which we denote $f(\boldsymbol{x}, \theta)$, is shared between the target and sensitive representation. The output of this shared encoder is fed through two separate encoders

78 $q_{\theta_T}(\boldsymbol{z}_T|\boldsymbol{x})$ and $q_{\theta_S}(\boldsymbol{z}_S|\boldsymbol{x})$, which each output a distribution in the latent space, and from which the target and sensitive
79 representations are sampled, respectively. Here, $\theta_T$ and $\theta_S$ denote the sets of trainable parameters for either encoder,
80 and include the parameters for the shared encoder, which can be found by $\theta = \theta_T \cap \theta_S$.

81 The target label $\hat{\boldsymbol{y}}$ is then predicted by a target discriminator $q_{\phi_t}(\boldsymbol{y}|\boldsymbol{z}_T)$, based on the target representation $\boldsymbol{z}_T$. Similarly,
82 the sensitive label $\hat{\boldsymbol{s}}$ is predicted by a sensitive discriminator $q_{\phi_S}(\boldsymbol{s}|\boldsymbol{z}_S)$, based on $\boldsymbol{z}_S$. The encoder and discriminators
83 are trained in supervised fashion to minimize the following losses, which we call the representation losses:

$$\mathcal{L}_T(\theta_T, \phi_T) = KL(p(\boldsymbol{y}|\boldsymbol{x}) \,\|\, q_{\phi_t}(\hat{\boldsymbol{y}}|\boldsymbol{z}_T)) \tag{1}$$

$$\mathcal{L}_S(\theta_S^*, \phi_S) = KL(p(\boldsymbol{s}|\boldsymbol{x}) \,\|\, q_{\phi_S}(\hat{\boldsymbol{s}}|\boldsymbol{z}_S)) \tag{2}$$

84 Here $\theta_S^* = \theta_S \backslash \theta$. These losses are effectively equal to the cross-entropy between the predicted values for the targets
85 and sensitive attributes and their actual values. Note that by backpropagating our sensitive representation loss through
86 $\theta_S^*$, the shared parameters $\theta$ are prevented from being updated twice.

87 To ensure that no sensitive information can leak into the target representation, the entropy of the sensitive attribute given
88 the target representation is maximized, following Roy and Boddeti [2019a], Sarhan et al. [2020]. This is achieved by
89 minimizing

$$\mathcal{L}_E(\phi_S, \theta_T) = KL(q_{\phi_S}(\boldsymbol{s}|\boldsymbol{z}_T) \,\|\, \mathcal{U}(\boldsymbol{s})) \tag{3}$$

90 Note that $\boldsymbol{z}_T$ can be used as the condition (or input) for $q_{\phi_S}$ because $\boldsymbol{z}_T$ has the same dimensionality as $\boldsymbol{z}_S$ (they are
91 orthogonal in the same space).

92 Last, we want to ensure that there is some level of independence between the two representations. Ideally, the posterior
93 $p(\boldsymbol{z}_T|\boldsymbol{x})$ should be statistically independent of $p(\boldsymbol{z}_S|\boldsymbol{x})$. Following Sarhan et al. [2020], this independence requirement
94 is relaxed to the enforcing of two properties: one, a disentanglement property (i.e. independence across dimensions
95 within a representation), and two, orthogonality between the two representations. To enforce these properties, the
96 aforementioned posteriors need to be *estimated* (as they are intractable) using variational inference [Kingma and
97 Welling, 2014]. The encoder network is be similar to the encoder of a Variational Auto-Encoder (VAE) model [Kingma
98 and Welling, 2013], in that it outputs the means ($\boldsymbol{\mu}_T, \boldsymbol{\mu}_S$) and covariance matrix diagonals ($\text{diag}(\boldsymbol{\sigma}_T), \text{diag}(\boldsymbol{\sigma}_S)$) for
99 both latent distributions. Disentanglement is enforced by only computing the diagonals of these covariance matrices,
100 while orthogonality is enforced by minimizing the KL divergence between each latent distribution with its prior, where
101 the priors $p(\boldsymbol{z}_T)$ and $p(\boldsymbol{z}_S)$ are initialized with orthogonal means. Two loss terms are introduced to achieve this
102 minimization: $\mathcal{L}_{z_T}(\theta_T) = KL(q_{\theta_T}(\boldsymbol{z}_T|\boldsymbol{x}) \,\|\, p(\boldsymbol{z}_T))$ and $\mathcal{L}_{z_S}(\theta_S) = KL(q_{\theta_S}(\boldsymbol{z}_S|\boldsymbol{x}) \,\|\, p(\boldsymbol{z}_S))$

103 Here $q_{\theta_T}(\boldsymbol{z}_T|\boldsymbol{x}) = \mathcal{N}(\boldsymbol{z}_T|\boldsymbol{\mu}_T, \text{diag}(\boldsymbol{\sigma}_T^2))$ and $q_{\theta_S}(\boldsymbol{z}_S|\boldsymbol{x}) = \mathcal{N}(\boldsymbol{z}_S|\boldsymbol{\mu}_S, \text{diag}(\boldsymbol{\sigma}_S^2))$.

104 These two loss terms are combined into a single term, which is called the *Orthogonal Disentangled (OD)* loss:

$$\mathcal{L}_{OD}(\theta_T, \theta_S) = \mathcal{L}_{z_T}(\theta_T) + \mathcal{L}_{z_S}(\theta_S) \tag{4}$$

105 The re-parameterization trick [Kingma and Welling, 2013] is used to sample from the approximated posterior distribution
106 in order to obtain the latent representations, which can then be fed to the respective discriminators.

107 All of the aforementioned individual loss terms are derived in more detail in Appendix A. They are combined into a
108 single loss term, and in doing so, we arrive at the following objective:

$$\underset{\theta_T, \theta_S, \phi_T, \phi_S}{argmin} \; \mathcal{L}_T(\theta_T, \phi_T) + \mathcal{L}_S(\theta_{S^*}, \phi_S) + \lambda_E \mathcal{L}_E(\theta_T, \phi_S) + \lambda_{OD} \mathcal{L}_{OD}(\phi_T, \phi_S) \tag{5}$$

109 Here $\lambda_{OD}$ and $\lambda_E$ are loss weights which determine the relative importance of the OD loss and the entropy loss,
110 respectively. Additionally, two decay parameters $\gamma_{OD}$ and $\gamma_E$ are introduced which enable changing the weights of
111 these two losses while training. The loss weights at epoch $t$ during training are calculated as follows:

$$\lambda_{OD}^{(t)} = \lambda_{OD}^{(0)} \gamma_{OD}^{t/t_s} \tag{6}$$

$$\lambda_E^{(t)} = \lambda_E^{(0)} \gamma_E^{t/t_s} \tag{7}$$

112 Here $t_s$ is the so-called *step-size* hyperparameter, and $\lambda_{OD}^{(0)}, \lambda_E^{(0)}$ are the initial loss weights. The entropy loss weight
113 will be computed in the same way. $\lambda_{OD}^{(0)}, \lambda_E^{(0)}, \gamma_{OD}, \gamma_E$ and $t_s$ are all hyperparameters.

## 3.2 Datasets

In order to reproduce the results obtained by Sarhan et al. [2020] it was necessary to apply the model to five datasets. Below, we outline some basic properties of the datasets and we explain the sensitive and target attributes that are to be modeled. For detailed information about the datasets such as train/test splits, number of samples, and input dimensions, we refer to Table 3 in the Appendix.

**Tabular data**

The Adult and German dataset were obtained from the UCI repository [Dua and Graff, 2017]. Both of these datasets contain census data, and include categorical and continuous attributes which contain information about the person's gender, education, and occupation. For both datasets, preprocessing consisted of representing categorical columns in a one-hot encoding, where missing values were explicitly encoded as a separate category, while continuous variables were left unchanged.

For the Adult dataset, the task is to predict whether a persons income exceeds $50,000$, and the sensitive attribute is binary gender. For the German dataset, the task is to classify rows as having good or bad credit risk. Similar to the Adult dataset, the sensitive attribute is binary gender.

**YaleB data**

The Extended YaleB dataset was collected from the University of Toronto computer science department website [Georghiades et al., 2000]. Specifically, the 'Cropped' version of the dataset was used [Lee et al., 2005], which contains grayscale images of 38 human faces under different illumination conditions. The task is to identify to which of the 38 humans an image corresponds.

For this dataset, the reference work mentioned that the sensitive attribute was constructed by clustering the illumination conditions into five classes loosely corresponding to top left, bottom left, top right, bottom right, and center. However, the reference work (as well as other works to which it referenced) did not explicitly specify which illumination conditions had been assigned to which cluster, and thus, in our experiments, we have manually constructed these clusters. The explicit clusters we defined can be found in the Appendix. It should be noted that our majority class is not in line with the paper by Sarhan et al., as it is mentioned by them that a majority class classifier could attain 50% accuracy on their YaleB dataset, while this is around 35% in our case. Unfortunately, we were unable to find sufficient information to be able to replicate the ratios mentioned in the reference paper, and we presume that this difference is due to the fact that our clusters have been defined in a different way.

Our training dataset comprised of 190 images corresponding to one lighting position from each cluster, following [Sarhan et al., 2020, Louizos et al., 2015]. It is important to note that our testing dataset contained 2243 images, while the testing set in the reference work contained only 1096. The reason for this discrepancy is unclear, as we used the referenced version of the dataset, and found no mention of the omission of certain images in the reference paper.

**CIFAR data**

The CIFAR-10 and CIFAR-100 datasets were also collected from the University of Toronto computer science department website [Georghiades et al., 2000]. CIFAR-10 consists of colour images that are divided into 10 classes such as airplane, automobile and bird. Following Roy and Boddeti [2019a] and Sarhan et al. [2020], a new target attribute is constructed which denotes whether the subject of the image is alive or not (i.e., a binary class), and the sensitive attribute is defined as the original class of the image.

The CIFAR-100 dataset is similar to CIFAR-10, except that images are categorized as one of 100 total fine classes. The dataset also provides 20 coarse classes, which are a partition of the 100 fine classes, and which cluster similar fine classes into one category. As an example, fine classes 'beaver', 'dolphin' and 'otter' all belong to the coarse class 'aquatic mammals' (c.f. [Roy and Boddeti, 2019b]). Following Sarhan et al. [2020], the coarse classes are used as the target attributes, while the fine classes are used as the sensitive attribute.

## 3.3 Implementation details

Following the paper of Sarhan et al. [2020], we implement the following networks for the several datasets. Note that, for every MLP mentioned below, ReLU's are used as (non-final) activation functions. For the CIFAR-10 and CIFAR-100 tasks, the encoder used was the ResNet-18 architecture [He et al., 2016].

### 3.4 Hyperparameters

Most of the values used for the hyperparameters were taken directly from the reference work, or its supplement provided by Sarhan et al. However, optimal values for some hyperparameters were not reported, and thus, we empirically set these to values that seemed to result in satisfactory performance. We discuss which hyperparameters we were missing in Section 5, and report all hyperparameters that we used in the Appendix.

### 3.5 Experimental setup and code

**Setup Reproducibility**

Our implementation and instructions to run the code are available at https://github.com/paulodder/fact2021. The repository contains a folder `scripts` that contains all the scripts necessary to perform several tasks. All instructions for setting up are in the README and instructions for reproducing any of the numbers or figures reported in this text can be found in `produce_results.pdf` in the aforementioned repository.

**Evaluation**

Evaluation of the embeddings learned by our model is non-trivial, as it must be assessed whether the embeddings adequately represent the data for the downstream task (e.g. classification of target attribute), while it also must be verified that the embeddings contain no sensitive information. In order to quantitatively evaluate our model after completing training, two classifiers are trained. These classifiers use the test data which is embedded using our trained model in the target space.

The first classifier, known as the *target predictor*, is trained to predict the target label from the target embeddings. In accordance with the reference paper, we evaluated the target predictor using the accuracy metric. The details of the target predictors used are reported in Table 7 in the Appendix. It is desirable that the target predictor performs as well as possible, as this means that the target embeddings appropriately embed the information necessary for downstream tasks.

The second classifier, known as the *sensitive predictor*, is trained to predict the sensitive attribute from the target representation. For the sensitive predictor, the exact same architecture and hyperparameters are used as for the sensitive discriminator. It is desirable that this classifier performs poorly, as we would like there to be no information pertaining to the sensitive attribute in our target embedding. As such, we would like the model to be as close to a 'majority classifier' as possible, where the model is forced to simply predict the majority label for each data row as it has no meaningful information with which to make a prediction about the sensitive attribute. Again, accuracy is used as evaluation metric.

**Additional avenues of exploration**

For the sake of completeness, we briefly report alternatives that were explored but did not yield improved results, and were therefore abandoned. None of the features described below were used to generate results.

In order to select the best performing model to evaluate, two independent selection mechanisms were implemented, but not used in the final experiments. (1) We attempted to select the best iteration of the proposed model (over all epochs) by keeping track of the version in which performance was best. We first defined performance as train target accuracy (higher is better). Later, to also take into account the extent of sensitive information leakage in the target representations, we also included the accuracy of predicting sensitive attributes based on target representations. (2) In order to make the evaluative target and sensitive predictors less dependent on the number of epochs they are trained for (which was a hyperparameter not disclosed by the reference paper, while overfitting might lead to a severe reduction in the reported results), we attempted to track all iterations of these predictors, perform evaluation using all of the iterations, and return the evaluation which had the highest performance. Here, performance was defined as test target accuracy. However, this augmentation was discarded as we were unsure whether this was implemented correctly, as results did not improve (even though it should, in theory).

For YaleB, various model architectures were implemented in an attempt to amend performance on this dataset. We experimented with variations in the dimensionality and number of hidden layers of the encoder and discriminators, activation functions (specifically, we tried `Tanh`), and the hyperparameters learning rate, max epochs, batch size, $\lambda_{OD}$, $\lambda_E$, $\gamma_{OD}$, and $\gamma_E$.

For CIFAR-10 and CIFAR-100, we experimented with freezing the ResNet-18 encoder (with the exception of the final, Linear layer, which was reinitialized), but despite faster training, the model's performance did not increase.

Table 1: The average run-time for each of the five datasets and their configurations.

| Dataset | Adult | German | YaleB | CIFAR-10 | CIFAR-100 | Total |
|---|---|---|---|---|---|---|
| Average run-time (min.) | 0.8 | 0.22 | 2 | 11 | 19 | 62 |
| Number of epochs | 2 | 15 | 30 | 30 | 55 | - |

## 3.6 Computational Requirements

We used Google Colab Pro to train our models, which supplies one `Tesla V100-SXM2-16GB` GPU, and two `Intel(R) Xeon(R) CPU @ 2.00GHz` CPUs. Average run-times are specified in Table 1. In order to train all models over various seeds for all results, this would be the estimated required run-time:

$$(3 * 62) + (5 * 5 * 62) + (2 * 8^2 * 5 * 0.8) = 4,296 \text{ minutes}$$

# 4 Results

To judge the reproducibility of the model proposed by Sarhan et al. [2020], we compare their results with those results we were able to attain using our implementation. First, a comparison is made between their and our target and sensitive accuracy attained by training and evaluating the proposed model on each of the five datasets. Second, results of the ablative study are compared. Finally, results of the sensitive study are compared.

**CIFAR-10 and CIFAR-100**

Table 2: Results on CIFAR-10 and CIFAR-100 datasets

| | CIFAR-10 | | CIFAR-100 | |
|---|---|---|---|---|
| | Target Acc. ↑ | Sensitive Acc. ↓ | Target Acc. ↑ | Sensitive Acc. ↓ |
| Sarhan et al. | 0.9725 | 0.1907 | 0.7074 | 0.1447 |
| Ours | 0.9582 | 0.3462 | 0.0500 | 0.0100 |

While we have been able to reproduce the CIFAR-10 target accuracy attained by Sarhan et al., the CIFAR-10 sensitive accuracy we attained is substantially higher than theirs, as displayed in Table 2. As for the CIFAR-100 dataset, our results strongly differed from those reported by Sarhan et al., as our model was not able to learn a representation that carried meaningful information, resulting in target and sensitive accuracies that are equal to accuracies attained by majority vote (see Table 2).

**Adult, YaleB, and German**

Note that for the following results, we focus on the comparison between performances of the proposed models. We have included a comparison of the alternative models in Figure 1 mainly to be able to investigate discrepancies in our reimplementation outside of the proposed method itself (e.g. significant differences in the dataset definition, pre-processing, et cetera).

Our results for Adult, as displayed in Figure 1, are similar to those obtained by Sarhan et al. [2020], with the only difference being a small increase in our sensitive accuracy with regards to theirs. As for German, we observe similar, yet not identical, target and sensitive accuracies. It should be noted that for specific runs during training (with specific random seeds), a target accuracy was obtained that was identical to the 76% reported by Sarhan et al.; however, over multiple runs, we obtained a lower average accuracy around 73% (see Figure 1). For YaleB, we were not able to reproduce the accuracies reported by Sarhan et al. Instead, our model achieved a lower target accuracy, and a sensitive accuracy which is further away from the majority label classifier, suggesting that our model's performance was worse than that of Sarhan et al.

**Ablative**

The results of our ablative study are shown in Figure 2, which can be compared with the ablative study of Sarhan et al. in Figure 5 in Appendix B. As a discussion of the potential implications of the various combinations explored in this ablative study forego the scope of this paper, we refer to Sarhan et al. [2020] for a detailed overview. The baseline measurement was omitted as it was unclear from the text what it entailed.

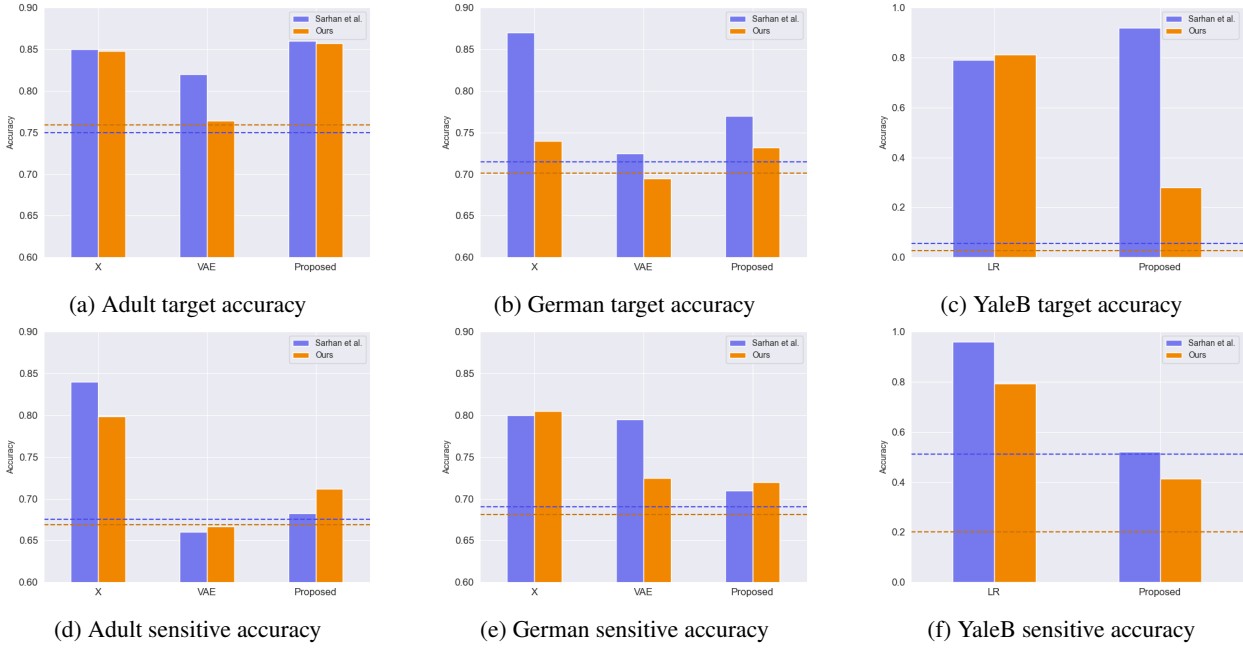

(a) Adult target accuracy    (b) German target accuracy    (c) YaleB target accuracy

(d) Adult sensitive accuracy    (e) German sensitive accuracy    (f) YaleB sensitive accuracy

Figure 1: Performance of the proposed model, together with majority label classifier (denoted by the horizontal dashed line) and various other models for Adult, German, and YaleB datasets, compared between Sarhan et al. and our reproduction. The bars denoted by X correspond to direct use of the input data for our target prediction. Furthermore, a VAE was trained on the Adult and German datasets using MSE loss as reconstruction loss, and the accuracies denoted with 'VAE' correspond to the performance achieved by target and sensitive predictors trained on these VAE embeddings as input features. For YaleB, Logistic Regression was also performed on the raw data to predict the sensitive and target attributes, whose performance is denoted by 'LR'.

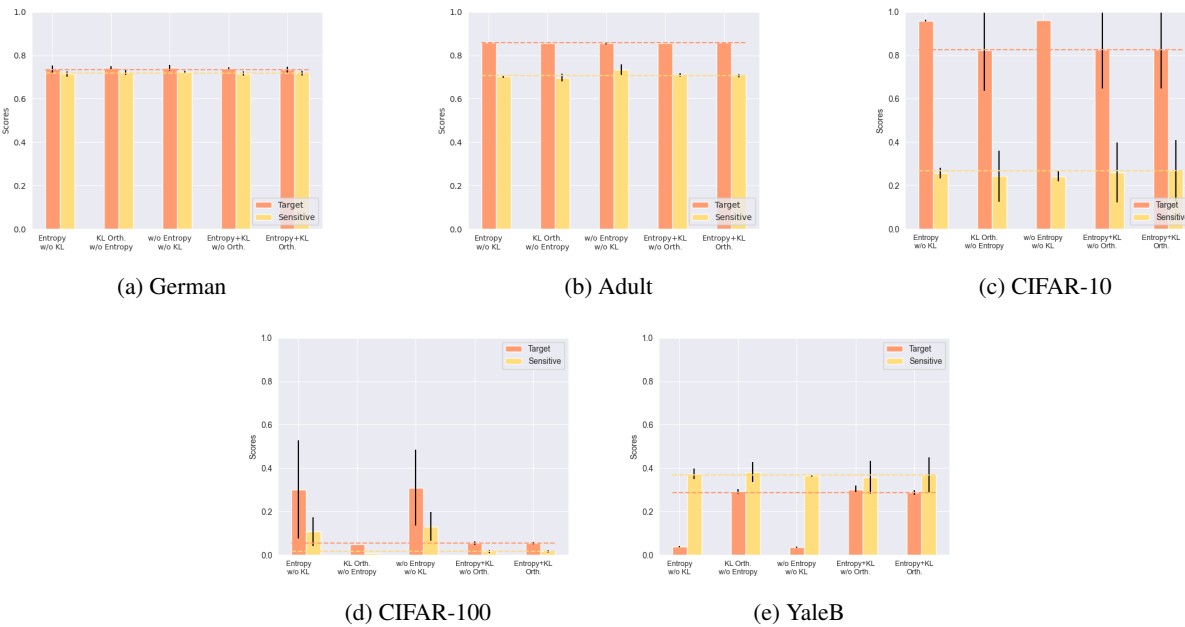

(a) German    (b) Adult    (c) CIFAR-10

(d) CIFAR-100    (e) YaleB

Figure 2: Target and sensitive accuracies of our model trained using various combinations of loss term components, results are averaged over 5 random seeds. Specifically, Entropy refers to the $\mathcal{L}_E$ component, Orth refers to the orthogonality constraint between the prior means, and KL refers to the $\mathcal{L}_{OD}$ component (c.f. Sarhan et al. [2020]). The horizontal dashed line denotes the accuracies attained when using the full loss (also displayed as the right-most set of bars in each subplot).

In comparison to Sarhan et al., for German, we see that varying loss components seems to have less impact on performance; for Adult, we see similar invariability for target accuracy but a lower impact on sensitive accuracy; for CIFAR-10, we observe a larger variance in performance over seeds and loss components; and lastly, CIFAR-100 and YaleB results are significantly different. In summary, our ablative study results generally do not exhibit the same patterns as those of Sarhan et al. This may, however, be attributed to our use of random seed averaging, a technique which was not mentioned in the reference paper.

**Sensitivity analysis on Adult**

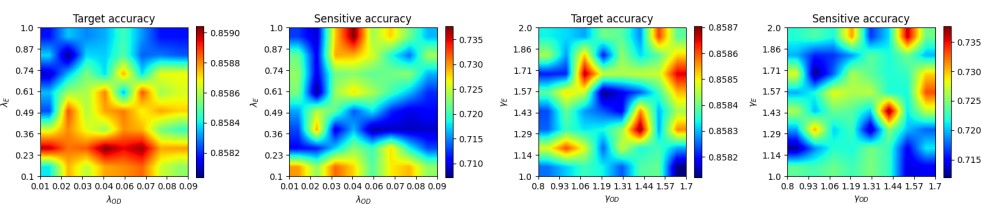

Figure 3: Target and sensitive accuracies when varying $\lambda_{OD}$ together with $\lambda_E$ (left), and when varying $\gamma_{OD}$ together with $\gamma_E$ (right).

The results of our sensitivity study are shown in Figure 3, which can be compared with the sensitivity study of Sarhan et al. in Figure 6 in Appendix B.

When comparing these sensitivity analyses, it can easily be observed that there is very little in common between the two. First off, there is, for each subfigure, a sizeable difference in the accuracy ranges. This difference is in line with differences encountered in Figures 1a and 1d. In addition, there is little similarity to be found in any of the accuracy landscapes displayed, with peaks and valleys located in different places. Keep in mind, however, that in both the reference and our sensitivity analyses, the accuracy ranges are rather small. Finally, in the reference sensitivity analysis, these landscapes are smooth, while this is not reflected in our sensitivity analysis. Note that the smoothness of the reference sensitivity analysis might be visually exaggerated due to a relatively low number of coordinate samples compared to ours.

# 5 Discussion

The main claim of the original authors is as follows: by disentangling the latent representation of a data sample into two subspaces that are orthogonal to each other, as well as training the model using a loss function that encourages it to encode sensitive information into one of these subspaces, and meaningful information for the task at hand into the other of those subspaces, it is possible to create meaningful representations that do not contain any information from which a protected, or sensitive, attribute can be inferred.

In order for our results to support this claim, they would need to show that the proposed model is able to create representations that perform well on the target task (i.e. attains a high target accuracy), while it performs poorly in the inference of the sensitive attribute using the target representation (i.e. attains a sensitive accuracy close to the accuracy of majority voting). When looking at our results, we observe that this is indeed the case for the German dataset. However, for the Adult and CIFAR-10 datasets, the attained sensitive accuracy is substantially higher than the majority vote baseline, and for the CIFAR-100 and YaleB datasets, the model does not achieve a satisfactory performance in terms of target accuracy. As for the YaleB dataset, this difference might be caused by a deviation in our perusal and/or construction of the dataset resulting from ambiguity in the reference paper; for the CIFAR-100 dataset, however, we are confident in the correspondence between our setup and that of the reference paper, and therefore question the reproducibility of that particular task.

In summary, results from four out of five datasets do not appear to support the original claim of the authors. Furthermore, those patterns that the authors observe in their ablative studies are generally not reproduced in our own ablative studies. Based on these observations, we can state that there is a discrepancy between our results and the original results from Sarhan et al. [2020]. Thus, when considering the large effort undertaken in this research to minutely re-implement their proposed method, we conclude that the original paper is relatively difficult to reproduce, and can in fact not be reproduced based solely on its contents.

### 5.1 What was easy

We experienced the theoretical part of the paper to be especially well-structured and thought out. The use of two types of representations and notions of disentanglement and orthogonality makes a lot of sense intuitively. Additionally, all loss terms are well described and were therefore easy to implement.

### 5.2 What was difficult

**Performance fluctuations and training instability**   One of the issues we ran into is that for these models training seems to be unstable, which is evident from the high fluctuation in performance when we vary the random seed or the number of maximum epochs. This is not addressed in the paper and therefore there is no information on how to deal with it. To add to this, it was unclear what trade-off between target and sensitive accuracy was used by the authors to select the best model during training. This trade-off ultimately determines which model is selected for testing which can have a large influence on performance.

**Implementation**   There were a few unclear aspects of the model implementation that we resolved either by making a choice that seemed logical to us, or through contacting the original author. For example, there was limited information on how certain losses were backpropagated with a shared encoder network. Besides this, the implementation of the decay the two $\lambda$ parameters was not clearly reported. These issues were both resolved in contact with the authors.

**Hyperparameters**   The amount of epochs that the model was trained was not reported in either the paper or its supplementary material. This was quite an important value given that no explicit stopping criterion was mentioned either. In correspondence with Sarhan et al., we were able to set values for the $t_s$ (stepsize) hyperparameter that correspond to those used by the original team. Furthermore, amongst the not reported hyperparameters were those involved the training of the network-based target and sensitive predictors. These include the optimizer used, the learning rate, weight decay, amount of epochs as well as the nonlinearities, to name a few.

**Dataset details**   As mentioned in YaleB paragraph of the Datasets section we have made a number of assumptions about how to set up the classes corresponding to the sensitive attributes, which might have some influence on the performance of our approach for this datasets. We were unsure about some other details concerning the data as well. Namely, the type of data-normalization is not specified, and for the German dataset there is not a train-test split reported. However, these details were not as vital for reproduction as the aforementioned issue concerning the YaleB dataset.

### 5.3 Communication with original authors

We have had the pleasure of communicating with the original authors of the paper. This enabled us to learn the values of some additional hyperparameters, such as the stepsize $t_s$, as well as the dimensions of the latent representations for some datasets. Furthermore, we were able to accrue additional insight in some implementation details, such as how the loss weights $\lambda_{OS}$ and $\lambda_E$ are updated, and how the losses should be backpropagated when dealing with a shared encoder network. In addition, it was the intention of the authors to supply us with more information on the YaleB dataset specifically, but we were not able to receive said information before the submission deadline of our work. This hiccup notwithstanding, communication with the authors was especially pleasant and straightforward.

### 5.4 Our approach

Due to the large scope of the research performed in our reference paper, our approach was diverse from the start. Many different avenues were explored from the beginning, dataloaders for all of the datasets were implemented and we had quickly written code to produce many of the figures necessary to asses the reproducibility of the research. While this meant that we gained a better understanding of the models' performance and behaviour on all of the datasets and tasks from the beginning, it was complicated to work on all the tasks and datasets simultaneously and evenly throughout the team.

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

# Appendix

## A Loss terms derivations

**Representation loss**

The representation target loss can be computed as follows:

$$\mathcal{L}_T(\theta_T, \phi_T) = KL(p(\boldsymbol{y}|\boldsymbol{x}) \parallel q_{\phi_T}(\boldsymbol{y}|\boldsymbol{z}_T))$$
$$= -\sum_{\boldsymbol{y}} p(\boldsymbol{y}|\boldsymbol{x}) \log q_{\phi_T}(\boldsymbol{s}|\boldsymbol{z}_T) + \sum_{\boldsymbol{y}} p(\boldsymbol{y}|\boldsymbol{x}) \log p(\boldsymbol{y}|\boldsymbol{x}) \tag{8}$$

The second part of this expression solely depends on the true posterior of our data and hence does not depend on our neural network. Therefore, we drop it here. What remains is equal to the cross-entropy loss:

$$\mathcal{L}_T(\theta_T, \phi_T) = \sum_{\boldsymbol{y}} p(\boldsymbol{y}|\boldsymbol{x}) \log q_{\phi_T}(\boldsymbol{s}|\boldsymbol{z}_T) \tag{9}$$

This is the same as the cross-entropy loss over the output of the discriminator. The representation sensitive loss can be computed in similar fashion.

**Maximum Entropy loss**

We can compute the entropy loss as follows:

$$
\begin{aligned}
\mathcal{L}_E(\phi_S, \theta_T) &= KL(q_{\phi_S}(\boldsymbol{s}|\boldsymbol{z}_T) \,\|\, \mathcal{U}(\boldsymbol{s})) \\
&= \sum_{\boldsymbol{s}} q_{\phi_S}(\boldsymbol{s}|\boldsymbol{z}_T) \log q_{\phi_S}(\boldsymbol{s}|\boldsymbol{z}_T) - \sum_{\boldsymbol{s}} q_{\phi_S}(\boldsymbol{s}|\boldsymbol{z}_T) \log \mathcal{U}(\boldsymbol{s}) \\
&= \sum_{\boldsymbol{s}} q_{\phi_S}(\boldsymbol{s}|\boldsymbol{z}_T) \log q_{\phi_S}(\boldsymbol{s}|\boldsymbol{z}_T) - \log \frac{1}{m} \sum_{\boldsymbol{s}} q_{\phi_S}(\boldsymbol{s}|\boldsymbol{z}_T) \\
&= \sum_{\boldsymbol{s}} q_{\phi_S}(\boldsymbol{s}|\boldsymbol{z}_T) \log q_{\phi_S}(\boldsymbol{s}|\boldsymbol{z}_T) + \log m
\end{aligned}
\tag{10}
$$

The second term is a constant and will be the same for every loss no matter the network, hence we drop it:

$$
\mathcal{L}_E(\phi_S, \theta_T) = \sum_{\boldsymbol{s}} q_{\phi_S}(\boldsymbol{s}|\boldsymbol{z}_T) \log q_{\phi_S}(\boldsymbol{s}|\boldsymbol{z}_T)
\tag{11}
$$

Note that by dropping the last term, the entropy loss will always be negative.

**Orthogonal-Disentangled loss**

We can write out the OD target loss as follows,

$$
\begin{aligned}
\mathcal{L}_{\boldsymbol{z}_T}(\theta_T) &= KL(q_{\theta_T}(\boldsymbol{z}_T|\boldsymbol{x}) \,\|\, p(\boldsymbol{z}_T)) \\
&= -\sum_{i=1}^{d_T} KL(q_{\theta_T} z_T^i|\boldsymbol{x}) \,\|\, p(z_T^i))
\end{aligned}
$$

because both the prior and the encoder posterior are independent Gaussian distributions, the KL divergence between the two is simply a sum over KL divergences between the univariate Gaussians $q_{\theta_T}(z_T^i|\boldsymbol{x})$ and $p(z_T^i)$.

One KL divergence terms can be computed as follows:

$$
\begin{aligned}
KL(q_{\theta_T}(z_T^i|\boldsymbol{x}) \,\|\, p(z_T^i)) &= -\int q_{\theta_T}(z_T^i|\boldsymbol{x}) \log \frac{q_{\theta_T}(z_T^i|\boldsymbol{x})}{p(z_T^i)} d\boldsymbol{x} \\
&= \frac{1}{2}\log(2\pi\sigma_{p_T}^i) + \frac{(\sigma_{q_T}^i)^2(\mu_{q_T}^i - \mu_{p_T}^i)^2}{2(\sigma_{p_T}^i)^2} - \frac{1}{2}(1 + \log 2\pi(\sigma_{q_T}^i)^2) \\
&= \log \frac{\sigma_{p_T}^i}{\sigma_{q_T}^i} + \frac{(\sigma_{q_T}^i)^2(\mu_{q_T}^i - \mu_{p_T}^i)^2}{2(\sigma_{p_T}^i)^2} - \frac{1}{2}
\end{aligned}
\tag{12}
$$

In practice, we will compute the element-wise KL divergence between the prior and posterior and sum over the result. The OD losses therefore require the output *means* and *variances* of the encoder network and the *prior distributions* of the latent variable. The OD sensitive loss can be computed in a similar way.

**B Dataset details**

Table 3: Details concerning the several datasets we used. Here MV target and MV sensitive correspond to how much percent of the data belongs to the biggest target and sensitive class respectively. The input size corresponds to the amount of features in the case of the tabular data and for the picture dimensions of the visual data.

| | sample amount | train/test split | input size | MV target | MV sensitive |
|---|---|---|---|---|---|
| Adult | $48,842$ | $2:1$ | $108$ | $75\%$ | $67\%$ |
| German | $1000$ | $4:1$ | $61$ | $68\%$ | $70\%$ |
| YaleB | $2433$ | $190:2243$ | $192 \times 168$ | $2.7\%$ | $35.6\%$ |
| CIFAR-10 | $60,000$ | $5:1$ | $3 \times 32 \times 32$ | $60\%$ | $10\%$ |
| CIFAR-100 | $60,000$ | $5:1$ | $3 \times 32 \times 32$ | $5\%$ | $1\%$ |

 **YaleB pre-processing**

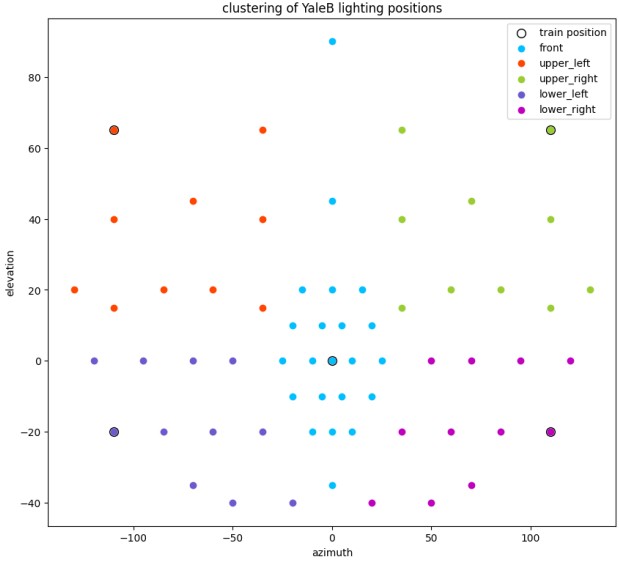

Figure 4: Definitions of YaleB sensitive attributes, which are a clustering of lighting positions, which are defined by an elevation and an azimuth.

In order to construct the sensitive attributes for the YaleB dataset, we define a five-class clustering for the lighting positions, which corresponds to a five-class sensitive attribute. These clusters, as well as the lighting positions that are selected for the train partition, are displayed in Figure 4.

## C Hyperparameters

The hyperparameters that we used for our reported results can be found in Table 4 and 5. Note that for all experiments we used the Adam optimizer [Kingma and Ba, 2014].

Table 4: Hyperparameters that we used in our experiments for the various datasets. For the CIFAR datasets, the first number of the learning rate and weight decays refers to the encoder network and the second to the discriminator network.

|  | Learning Rate | Weight Decay | Batch Size | Max. Epochs |
|---|---|---|---|---|
| Adult | $10^{-3}$ | $5 \times 10^{-4}$ | 64 | 2 |
| German | $10^{-3}$ | $5 \times 10^{-4}$ | 64 | 15 |
| YaleB | $10^{-4}$ | $5 \times 10^{-2}$ | 64 | 30 |
| CIFAR-10 | $10^{-4}, 10^{-2}$ | $10^{-2}, 10^{-3}$ | 128 | 30 |
| CIFAR-100 | $10^{-4}, 10^{-2}$ | $10^{-2}, 10^{-3}$ | 128 | 80 |

Table 5: The $\lambda_{OD}, \lambda_E, \gamma_{OD}$ and $\gamma_E$ used for every dataset.

|  | $\lambda_{OD}$ | $\lambda_E$ | $\gamma_{OD}$ | $\gamma_E$ |
|---|---|---|---|---|
| Adult | 0.037 | 0.55 | 0.8 | 1.66 |
| German | 0.01 | 1.0 | 1.4 | 2.0 |
| YaleB | 0.037 | 1.0 | 1.1 | 2.0 |
| CIFAR-10 | 0.063 | 1.0 | 1.7 | 1.0 |
| CIFAR-100 | 0.0325 | 0.1 | 1.2 | 1.67 |

## D Encoder and Discriminator details

See Table 6 for the specific implementation details regarding the encoder and discriminator models used.

Table 6: Encoder and discriminator implementation details.

| | Encoder | | | Discriminator | |
|---|---|---|---|---|---|
| | Network Type | Hidden Dims | Latent Dim | Network type | Hidden Dims |
| Adult & German | MLP | 64 | 2 | MLP | 64, 64 |
| YaleB | MLP | 100 | 100 | MLP | 100, 100 |
| CIFAR | ResNet-18 | - | 128 | MLP | 256, 128 |

## E Target predictor details

We have reported the architectures and hyperparameters of the target predictor networks in Table 7. We used the Adam optimizer [Kingma and Ba, 2014] to optimize all MLP-based predictor networks.

Table 7: Details of the target predictor network per dataset.

| | Network Type | Hidden Dims | Learning Rate | Weight Decay |
|---|---|---|---|---|
| Tabular | Logistic Regression | - | - | - |
| YaleB | MLP | 100 | $10^{-3}$ | 0 |
| CIFAR | MLP | 256, 128 | $10^{-3}$ | 0 |

## F Ablative and sensitive study results in Sarhan et al. (2020)

For ease of comparison, we include two Figures from the reference paper. All rights for Figures 5 and 6 reserved by Sarhan et al.

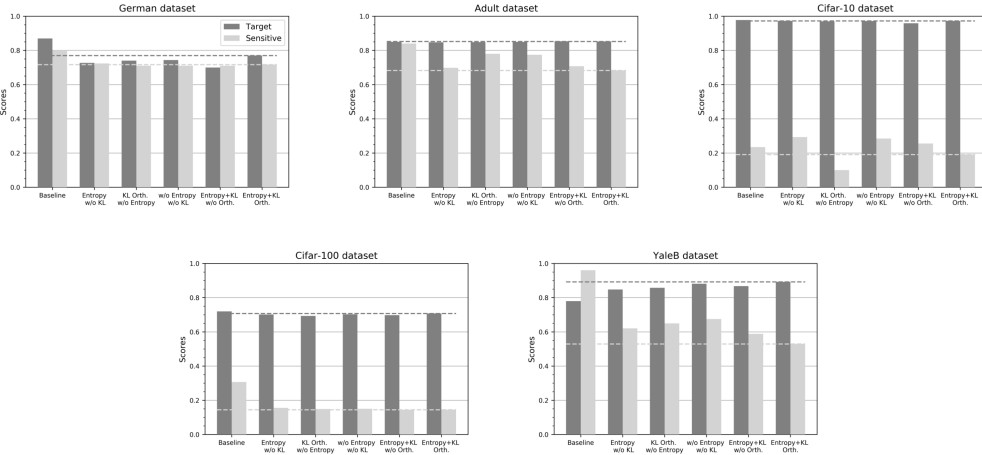

Figure 5: Figure 3 from Sarhan et al. [2020], with original caption: *Ablative study. Dark gray and light gray dashed lines represent the accuracy results on the target and sensitive task respectively for the "Entropy + KL Orth." model.*

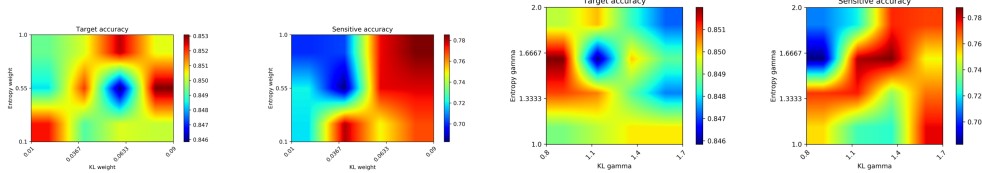

Figure 6: Figure 5 from Sarhan et al. [2020], with original caption: *Sensitivity analysis on the Adult dataset*

