# OpenReview forum: "Reimplementing Fairness by Learning Orthogonal Disentangled Representations"
_ML_Reproducibility_Challenge/2020 — Reject_

### Official Review · AnonReviewer3 · 2021-02-22
**A good reimplementation**

**Rating:** 8
**Confidence:** 4

**Review:**

The authors of this report provide a complete open-source reimplementation of the original method as well as the figures provided in the original paper. They find some discrepancies and variance in the results. The authors do not go beyond the original content of the paper in terms of understanding the method; they do superficially report some unsuccessful attempts to improve it.

Format: the authors correctly follow the format required for the challenge.
Scope: the authors (attempt to) reproduce all the results from the paper
Code: at a glance the provided code appears complete. It was written from scratch since no code is provided in the Original Paper
Communication & hyperparameters: the authors have done their due diligence
Ablation: the authors perform the same ablation study as in the OP, finding some differences, but do not report further ablative modifications. They also perform a similar sensitivity analysis.
Discussion: the authors make appropriate discussions and remarks on reproducibility

Comments:
- there seems to be a typo in eq (4), \lambda was replaced by some text
- broken reference on line 171
- Figure 1: the labels of the plots are pretty small, in academic papers labels should have the same font size than the main text
- “We first defined performance as train target accuracy, [then test]”, this is a nice insight, but not the correct way to do this. If using the training data, then you are only measuring memorization, if using the test data, then you’ve “cheated” and your test data is no longer test data, and no longer gives you an unbiased estimate of the performance of your method. The correct way to do this would be to split your training data in two, i.e. create a validation set, and use this validation set solely to determine early stopping.
- missing Figure reference on lines 224 and 236
- the authors critique the OP for not having the same number of random seeds per hyperparameter setting. Is this referenced somewhere in the OP? Or is this from an email exchange? These numbers should be made explicit.
- Figure 3 & sensitivity analysis: the authors note that “there is very little similarity to be found in any of the accuracy landscapes”. Experiments have natural variance, especially since different random seeds are used (and small code differences probably still remain). Considering the closeness (there isn’t much difference between 0.71 and 0.73) of the bounds of the colorbars it would be quite unexpected for the same peaks and valleys to show up. This shows that this method is robust to these two hyperparameter choices, or alternatively, that these two hyperparameters do not influence the results.
- stylistic comment: the authors use the “we” voice in a lot of contexts, this makes it hard to distinguish between what is something that they contribute (e.g. novel measures, novel hyperparameters or other choices) and something done by the original authors. Alternatively, the passive voice can be used, for example:
  - “We combine these two loss terms into a single term” -> “These two loss terms are combined into a single term”
  - “after completing training, we train two classifiers” -> “after completing training, two classifiers are trained”
- on language & grammar: the text is very readable, but contains some typos and grammatical mistakes and would benefit from additional proofreading.


**Familiar With The Original Paper:**

I have not read the original paper

**Reproducibility Summary:**

Report has summary

---

### Official Review · AnonReviewer2 · 2021-02-27
**A solid reproduction effort, but not entirely convincing**

**Rating:** 6
**Confidence:** 4

**Review:**

This report presents an attempt at reproducing results from « Fairness by learning orthogonal disentangled representations » (Sarhan et al., 2020). In that paper, a representation learning algorithm is proposed to learn « fair » representations (i.e., representations from which sensitive attributes cannot be easily predicted) by learning two separate representations (one for the class of interest and one for sensitive attributes) that should be both disentangled and orthogonal to each other. This reproducibility work is only partially able to reproduce the original paper’s results, in part at least due to the lack of open source implementation and some missing experimental details.

The report’s authors clearly state their objective, explain what they have done (and what was easy/difficult), and compare their results to the original paper. This is a significant effort since they had to start from scratch, and they attempted to reproduce more than just the main results (they also include an ablation study and a sensitivity analysis). I also appreciate that there is a well documented open source repository to easily re-run these experiments.

My main concern is that it remains unclear why some of the results are so far off from the original paper. In particular, the YaleB and CIFAR-100 results are worrying, since the model is clearly unable to learn the task at hand. I would have expected the authors to dig deeper on that, for instance by doing more ablations / hyper-parameter search to at least identify which parts of the loss / model are responsible for this issue. Without such an analysis, there remains some doubt regarding the correctness of the implementation.

Another potential concern is that it is not clear from the descriptions of the datasets in 3.2 that the definitions of the sensitive attributes match what was done by Sarhan et al. on the YaleB and CIFAR datasets. Could you please clarify that point in the report?


Small remarks:
- On .81 should « based on z_T » be « based on z_S »?
- I find eq. 3 a bit confusing because q_phi_S is applied with z_T as input instead of z_S: how is that possible?
- In eq. 4 there is probably a missing \ in the Latex code
- l. 171 has a missing reference
- What are the horizontal dashed lines in the plots?
- References to figures in the Appendix seem broken (ex: l. 224, but there are more)
- l. 360: « We should check whether these are in fact the last hyperparameters we used » => was this sentence supposed to be included?


**Familiar With The Original Paper:**

I have not read the original paper

**Reproducibility Summary:**

Report has summary

---

### Decision · Program_Chairs · 2021-03-31

**Decision:**

Reject

**Comment:**

Good report, with implementation from scratch. However, ACs share the concern of reviewers regarding Yale-B and CIFAR100 results, which could potentially be an implementation issue. Thus, the AC's are unable to recommend this report to the journal.